# Screening Mammogram Classification with Prior Exams

**Jungkyu Park**[1]**, Jason Phang**[1]**, Yiqiu Shen**[1]**, Nan Wu**[1]**, S. Gene Kim**[2]**, Linda Moy**[2]**, Kyunghyun Cho**[1]**, Krzysztof J. Geras**[2,1]

[1] *Center for Data Science, New York University*

[2] *Department of Radiology, New York University School of Medicine*

## 1. Introduction

Screening mammography had been shown to significantly reduce the mortality rate for breast cancer (Kopans, 2002; Duffy et al., 2002a,b), the second leading cause of cancer-related deaths among women in the United States. However, there is a high rate of false positive recalls and biopsies associated with breast cancer screening. Among the 10–15% of women asked for recall, only 10–20% within that subset are recommended for biopsy. Among those biopsies, only 20–40% are diagnosed with cancer (Kopans, 2015).

Given the success of deep learning in computer vision, many deep neural network models have been applied to breast cancer screening (Ribli et al., 2018; Lotter et al., 2017; Geras et al., 2017; Wu et al., 2018, 2019a). Typically, these models operate on a single screening exam. However, radiologists often compare current mammograms to prior ones to make more informed diagnoses (Roelofs et al., 2007; Hayward et al., 2016). For instance, if a suspicious region grows in size or density over time, radiologists can be more confident that it is malignant. Conversely, if a suspicious region does not grow, then it is probably benign.

The goal of this work is to construct a model that can take advantage of prior exams in making a diagnosis. Concretely, we train models that take two screening exams as input, with each exam containing four images. For each corresponding image pair, the model produces predictions for the presence of benign or malignant findings in the more recent exam. An ensemble of such models achieves an AUC of 0.8664 for predicting malignancy in the screening population and 0.7987 for the subpopulation of the screening population that underwent biopsy, reducing the error rate of the corresponding baseline (Wu et al., 2019a).

## 2. Data

We use the NYU Breast Cancer Screening Dataset (Wu et al., 2019b) used in Wu et al. (2019a). The dataset consists of 229,426 exams, with each exam consisting of at least one image for the four standard views (L-CC, R-CC, L-MLO, R-MLO). We use the four binary labels corresponding to the presence of benign or malignant findings in the left or the right breasts. In this work, we consider only the subset of this dataset that includes patients for which prior exams are available. We define an *exam pair* to consist of a chronologically earlier and a later exam from the same patient, and an *image pair* to be the corresponding pair of images for the same view within the exam pair. For the training and validation set, we generate all combinations of such exam pairs. In the test set we only use pairs that involve the most recent exam of the patient as the later exam. Our dataset thus consists of 127,451 (respectively 25,111; 13,702) exam pairs from 43,013 (respectively 7,962; 7,600)

patients in training (respectively validation; test) set where 2,519 (respectively 393, 244) pairs had at least one biopsy performed. We refer to the population of patients who had at least one biopsy performed as *biopsied population*. Each image is cropped or padded to a fixed size of $2677 \times 1942$ pixels for CC view images and $2974 \times 1748$ for MLO view images.

### 2.1. Image Alignment

When a patient has multiple exams, the images for each view taken at different times can appear at different angles, sizes, or even different resolutions. We align the images within each pair before feeding them to our model in order to detect local changes without requiring the model to learn alignment. Concretely, we use two CNN models for geometric matching (Rocco et al., 2017), one trained using VGG (Simonyan and Zisserman, 2014) and one using ResNet-101 (He et al., 2016), for feature extraction. These models take a pair of images as input and output the parameters of an affine transformation to align the images from the prior exams to images from the recent exams. Using two transformation parameters from both models, we choose the one with better IoU of the nonzero masks of the registered source and target images (cf. Figure 1).

## 3. Comparison Models

We propose two architectures that incorporate information from pairs of images. The *Global-Compare* model applies the ResNets from the single-exam baseline model (image-only image-wise model from Wu et al. (2019a)) to both images and concatenates the two representations after global average pooling to obtain one representation per image pair. The *AlignLocalCompare* model concatenates image representations before global average pooling and applies an additional 1x1 convolutional layer that preserves the number of channels, followed by a ReLU activation function for local comparison. The

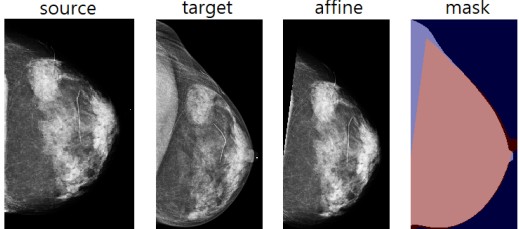

Figure 1: Alignment of image pairs. *Source* is the prior image, *target* is the current image, *affine* is registered *source* and *mask* represents the overlay of *target* and *affine*.

model architectures are shown in Figure 2. In both networks, we pass the resulting representation to a hidden layer and then to a softmax layer to obtain benign and malignant predictions for each image. Predicted probabilities for the same breast are averaged.

## 4. Experiments

We train the *GlobalCompare* model without aligning pairs of images, and the *AlignLocalCompare* with aligned image pairs. For each network, we follow Wu et al. (2019a) and construct an ensemble of five model instances. We load and freeze the weights of the ResNets from the respective five model copies in Wu et al., randomly initialize the weights of remaining layers, and train each model for 70 epochs. Each epoch consists of 2,519 pairs from the biopsied population and 2,519 randomly sampled from the rest of the population. We evaluate after every training epoch and choose the best weights based on the validation AUC on malignant prediction for the biopsied population. We report the test performance for each network using the ensemble of five model instances.

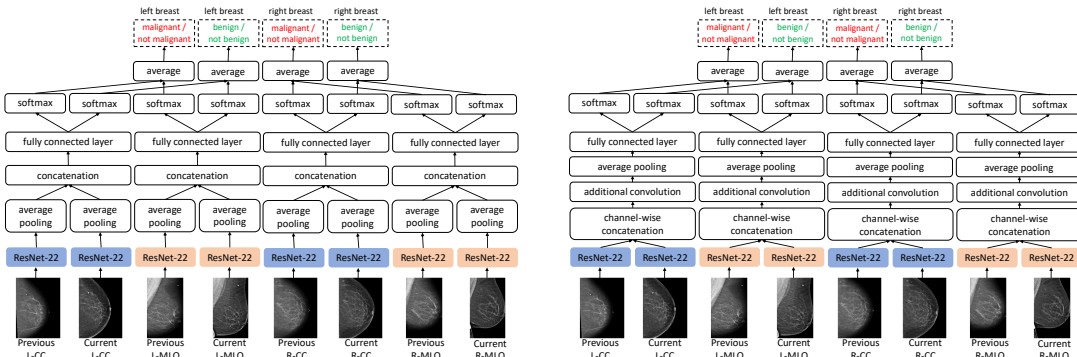

Figure 2: Architecture Diagrams. Left: *GlobalCompare*, right: *AlignLocalCompare*.

The results from all three models are shown in Table 1. *AlignLocalCompare* performs better for malignant prediction than the single-exam baseline and *GlobalCompare*, in both the biopsied and screening populations. We do not observe improvement for benign category–we speculate that this is because our model learns to focus on regions with significant changes, but not many changes accompany benign findings. In Figure 3, we visualize a few cases where the *AlignLocalCompare* is more confident in its prediction than the single-exam baseline (Wu et al., 2019a). For Figure 3(a), we observe that the malignant region did not exist in the prior exam. For Figure 3(b), we observe that the bright spot at the center already existed in the prior exam, and the model can be more sure that it is not malignant.

| | single (average of individual AUCs) | | 5x ensemble | |
|---|---|---|---|---|
| | malignant | benign | malignant | benign |
| **screening population** | | | | |
| single-exam baseline | 0.8368 (std 0.0126) | **0.7334** (std 0.0116) | 0.8442 | **0.7421** |
| *GlobalCompare* | 0.7871 (std 0.0359) | 0.6943 (std 0.0222) | 0.8065 | 0.7232 |
| *AlignLocalCompare* | **0.8419** (std 0.0211) | 0.7065 (std 0.0198) | **0.8664** | 0.7233 |
| **biopsied population** | | | | |
| single-exam baseline | 0.7548 (std 0.0123) | **0.6032** (std 0.0110) | 0.7596 | **0.6071** |
| *GlobalCompare* | 0.7214 (std 0.0430) | 0.5839 (std 0.0112) | 0.7421 | 0.5958 |
| *AlignLocalCompare* | **0.7761** (std 0.0235) | 0.5866 (std 0.0247) | **0.7987** | 0.5902 |

Table 1: Test AUCs calculated on subset with at least one prior exam. The larger variance of *Align-LocalCompare* contributes to its ensemble performing better.

Figure 3: Test examples where *AlignLocalCompare* performs better than the single-exam baseline. A breast with a malignant finding shown in (a) (malignant finding is highlighted with red) and one with a benign lesion shown in (b). *Align-LocalCompare* predicts malignancy with 0.97 probability for (a) and 0.04 for (b), whereas the baseline predicts 0.73 for (a) and 0.24 for (b). There is about a year gap between two exams for both patients.

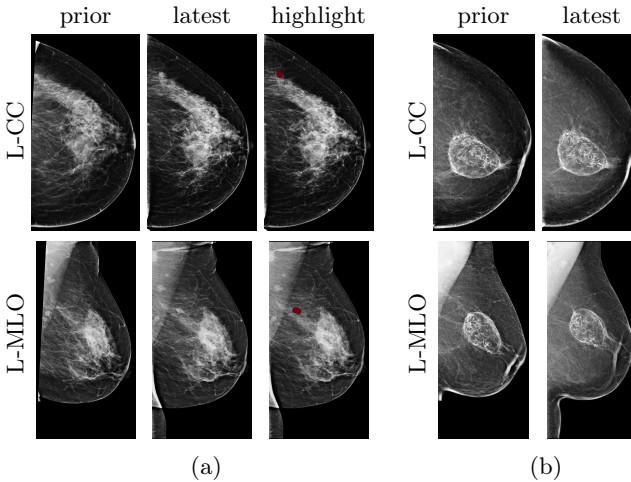

## Acknowledgments

We gratefully acknowledge the support of Nvidia Corporation with the donation of some of the GPUs used in this research. This work was supported in part by grants from the National Institutes of Health (R21CA225175 and P41EB017183). Jungkyu Park was supported by the Moore-Sloan Data Science Environment at New York University.

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
