# OpenReview forum: "Screening Mammogram Classification with Prior Exams"
_MIDL.io/2019/Conference/Abstract — MIDL Abstract 2019_

### Official Review · AnonReviewer2 · 2019-04-24
**Great idea for clinical practice**

**Rating:** 3
**Confidence:** 3

**Review:**

The desire of using previous results are strong now in the clinical world.
The author chose this important topic to work with.
Even the methodology is not new, this work pushes the great topic.
The abstract paper is well written with decent validation.

---

### Official Review · AnonReviewer1 · 2019-04-30
**A novel approach with through experiments**

**Rating:** 4
**Confidence:** 2

**Review:**

The abstract presents a model to classify mammography images as benign or malignant from multiple time steps. Images from different screening times are aligned by finding the parameters of the affine transformation. The paper presents two different models trained with and without aligned pairs to perform the classification task. Sufficient experiments are shown indicating the effectiveness of the method. The paper is very well written.

---

### Decision · Program_Chairs · 2019-05-06
**Acceptance Decision**

Accept